# Designing of Airspeed Measurement Method for UAVs Based on MEMS Pressure Sensors

**DOI:** 10.3390/s24175853

**Published:** 2024-09-09

**Authors:** Zhipeng Chen, Haojie Li, Hang Yu, Yuan Zhao, Jing Ma, Chuanhao Zhang, He Zhang

**Affiliations:** School of Mechanical Engineering, Nanjing University of Science and Technology, Nanjing 210094, China; czp@njust.edu.cn (Z.C.); yuanzhao@njust.edu.cn (Y.Z.); jing_ma@njust.edu.cn (J.M.); chuanhao.zhang@njust.edu.cn (C.Z.); hezhangz@njust.edu.cn (H.Z.)

**Keywords:** improved AR-SHAKF algorithm, airspeed measurement, UAVs, MEMS pressure sensors

## Abstract

Airspeed measurement is crucial for UAV control. To achieve accurate airspeed measurements for UAVs, this paper calculates airspeed data by measuring changes in air pressure and temperature. Based on this, a data processing method based on mechanical filtering and the improved AR-SHAKF algorithm is proposed to indirectly measure airspeed with high precision. In particular, a mathematical model for an airspeed measurement system was established, and an installation method for the pressure sensor was designed to measure the total pressure, static pressure, and temperature. Secondly, the measurement principle of the sensor was analyzed, and a metal tube was installed to act as a mechanical filter, particularly in cases where the aircraft has a significant impact on the gas flow field. Furthermore, a time series model was used to establish the sensor state equation and the initial noise values. It also enhanced the Sage–Husa adaptive filter to analyze the unavoidable error impact of initial noise values. By constraining the range of measurement noise, it achieved adaptive noise estimation. To validate the superiority of the proposed method, a low-complexity airspeed measurement device based on MEMS pressure sensors was designed. The results demonstrate that the airspeed measurement device and the designed velocity measurement method can effectively calculate airspeed with high measurement accuracy and strong interference resistance.

## 1. Introduction

Airspeed is a crucial parameter for controlling unmanned aerial vehicles (UAVs) during flight. It serves as a vital indicator for UAV control, enabling functions such as navigation, positioning, flight performance monitoring, maintaining flight stability, and enhancing flight safety. Common methods for measuring airspeed in UAVs include Pitot tube velocity measurements [1,2,3,4] and an integrated Global Positioning System (GPS) with a low-cost Inertial Measurement Unit (IMU) [5,6,7,8]. For small UAVs with lower cost and less stringent airspeed measurement requirements, GPS and IMU strategies can be expensive. Additionally, Pitot tubes can suffer from sensor measurement errors due to icing, dust, or particle blockages [9,10]. Large fixed-wing UAVs typically employ air data systems such as blades and porous Pitot tube probes. However, the airspeed measurement systems used in small UAVs have strict requirements regarding size, weight, power consumption, and cost. Therefore, there is an urgent need to design an airspeed measurement system using standard low-cost and high-integration sensors to achieve high-precision airspeed measurement for small UAVs.

Airspeed measurement systems rely on pressure and temperature sensors. Traditional pressure sensors have drawbacks, such as large size and high cost, which make them unsuitable for airspeed measurements in small UAVs.The development of Micro-Electro-Mechanical System (MEMS) technology has made pressure sensors cost-effective, highly accurate, and compact [11], making them suitable for small UAVs. While MEMS pressure sensors offer significant advantages, they also have their own set of challenges. A major issue is sensor noise, which can affect accuracy and, consequently, overall system performance. Reference [12] demonstrated through experiments that MEMS pressure sensors require the use of appropriate algorithms to fuse sensor data and improve data reliability. In [13], the Allan variance method was used to analyze different types of noise in a sensor. Their results indicated that random walk or white noise dominates in MEMS pressure sensors. In [14], a low-pass Butterworth filter was used to filter sensor data, and the algorithm was shown to effectively enhance the measurement accuracy of MEMS pressure sensors while exhibiting good noise resistance. In [15], various methods, such as weighted recursive averaging filtering, first-order lag software filtering, and recursive least squares filtering, have been applied to process MEMS pressure sensor data, all achieving certain levels of success. However, these methods do not provide a detailed noise modeling analysis of MEMS pressure sensors. In [16], MEMS pressure sensor noise was modeled as an exponential autocorrelation function with a time constant, demonstrating that sensor noise is time-dependent. In [17], the paper proposes an integrated control framework of Torque Vectoring (TV) and Active Front Steering (AFS) systems to ensure the stability of the vehicle’s lateral dynamics. The paper [18] proposes a hierarchical framework based on dual-model predictive control (MPC) to achieve energy savings while enhancing the processing stability of DDEV.

Considering that MEMS pressure sensor noise is time-dependent, classic time series analysis models such as autoregressive (AR), moving average (MA), and autoregressive moving average (ARMA) models are commonly used for modeling. Time series models are particularly suitable for MEMS inertial sensors and are commonly used for modeling gyro random noise [19,20,21]. In [22], an AR model was used to model MEMS IMUs, while in [23], an AR model was used for modeling quartz flexure gravity sensors. When using time series analysis methods to model MEMS sensors and obtain their state equations and initial parameters, the Kalman filter (KF) is often employed to denoise the sensor signals.

The standard KF requires accurate process noise and measurement noise information; otherwise, it can suffer from filter divergence [24]. The literature [25] proposes an adaptive fuzzy-KF model based on fuzzy criteria, which can achieve real-time updates of process noise and measurement noise. Reference [26] proposed an adaptive fuzzy-KF model based on fuzzy criteria, which enables real-time updates of process noise and measurement noise. Additionally, the Sage–Husa adaptive filter can also estimate noise statistical characteristics in real time but cannot guarantee the positive definiteness of the noise covariance matrix and the semipositive definiteness of the process noise covariance matrix [27]. Reference [28] introduced adaptive technology in an extended Kalman filter (EKF) by employing the Sage–Husa adaptive filter to estimate process and measurement noise covariances in real time. Reference [29] proposed innovative thresholds to enhance the filter’s response speed and employed a normalized cross-correlation method to improve tracking robustness. It achieved a simplified Sage–Husa adaptive Kalman filter (SHAKF) adaptive estimation that compressed the statistical characteristics of noise in tracking results. Reference [30] integrated the Sage–Husa adaptive filter algorithm and sequential Kalman filter (SKF) methods into a nonlinear cubature Kalman filter (CKF). This adaptively adjusts the measurement noise and maintains the positivity of the covariance matrix. The literature [31] proposes a Sage–Husa adaptive Kalman filtering-based pedestrian characteristic parameter update mechanism, which enhances step length estimation in pedestrian dead reckoning, resulting in improved accuracy. The literature [32] presents an improved Sage–Husa Adaptive Robust Kalman Filtering (MSHARKF) algorithm, which reduces random error noise to one ten-thousandth and one hundredth of the original data. In summary, the main issue with the Sage–Husa adaptive filter algorithm, as evident from the above references, is filter divergence caused by the uncertainty in filter noise.

The main contributions of this article are as follows:

(1) To address the excessive size and cost of conventional airspeed measurement devices, a mathematical model for airspeed measurement in small unmanned drones was established, and an airspeed measurement device was designed. The measurement device was designed with low cost in mind, utilizing two MEMS pressure sensors placed vertically and equipped with metal tubes to ensure airflow stability, thereby enhancing measurement accuracy.

(2) To address the issue of unknown initial noise values in standard KFs and the potential divergence of measurement noise, a time series model for the pressure sensor has been established. Based on static sensor data, an autoregressive third-order (AR(3)) model for the sensor was developed, and the sensor’s state equation and initial noise values were determined. Furthermore, this article analyzes the significant impact of initial noise values in the Sage–Husa filter, improving the Sage–Husa adaptive filtering algorithm by constraining the numerical magnitude of the measurement noise matrix to ensure filter convergence and achieve an adaptive estimation of measurement noise.

(3) Finally, wind tunnel tests show that the airspeed measurement device designed in this paper, along with the proposed adaptive AR-SHAKF algorithm, achieved significantly lower ME, MAE, and RMSE compared to the KF, demonstrating a notable improvement in noise reduction and more accurate airspeed calculation results.

The process of this paper is shown in Figure 1, and the structure is as follows. In Section 2, the principle of airspeed measurement and the MEMS air pressure sensor model are analyzed. In Section 3, the mechanical filtering structure of the airspeed measuring device is designed. In Section 4, the AR-SHAKF algorithm is improved. In Section 5, the wind tunnel experiments are presented and the results are analyzed. Finally, Section 6 presents the conclusions.

## 2. Mathematical Model of the Airspeed Measurement System

Airspeed measurement methods are based on the Bernoulli equation, and the relationship between air pressure and an unmanned aerial vehicle during flight is as follows:(1)$$p_{t}=p_{d}+p_{s}$$

In the equation, $p_{t}$ represents total pressure; total pressure refers to the sum of static pressure and dynamic pressure, typically measured perpendicular to the direction of airflow. $p_{d}$ represents the dynamic pressure; dynamic pressure refers to the pressure generated by the flow of gas. $p_{s}$ represents static pressure; static pressure refers to the pressure exerted by the flowing gas on the test point in the presence of stationary gas. The airspeed of an unmanned aerial vehicle is calculated as follows:(2)$$v_{UAV}=K\sqrt{2\left(p_{t}-p_{s}\right)/\rho_{s}}=K\sqrt{2p_{d}/\rho_{s}}$$
where *K* is the correction factor determined during the calibration of the Pitot-static system to account for the sensitivity of measurements to air temperature and pressure variations. $\rho_{s}$ represents air density, and it is calculated using the following formula [33]:(3)$$\rho_{s}=\frac{p_{s}M_{a}}{ZRT}\left[1-x_{v}\left(1-\frac{M_{v}}{M_{a}}\right)\right]$$
where $M_{a}$ is the molar mass of dry air. *Z* is the air compressibility factor. R is the molar gas constant. *T* is the thermodynamic temperature of the air. $x_{v}$ is the molar fraction of water vapor. $M_{v}$ is the molar mass of water. $M_{a}$ is the molar mass of dry air.

The factors that primarily affect air density during UAV flight are static pressure $p_{s}$ and temperature *T*. Equation (Equation 3) can be modified as follows:(4)$$\rho_{s}=b\frac{p_{s}}{T}$$
where $b=\frac{M_{a}}{ZR}\left[1-x_{v}\left(1-\frac{M_{v}}{M_{a}}\right)\right]$ is considered a constant. Substituting (4) into (2), we obtain
(5)$$v_{UAV}=K\sqrt{\frac{2T(p_{t}-p_{s})}{bp_{s}}}$$

### Modeling Sensor Noise Based on Time Series Models

The calculation of airspeed relies on the values of total pressure and static pressure, which are measured by sensors in the airspeed system. Sensor measurements inevitably introduce noise. In previous studies [34,35,36,37], KFs have often been employed to filter the noise from inertial sensors. However, before applying the filter, it is necessary to establish the sensor state equations and determine the noise covariance.

Based on previous analyses, it is evident that MEMS pressure sensor noise exhibits a time correlation. Therefore, this paper utilizes time series analysis methods to construct a state model for pressure sensors. Autoregressive (AR), moving average (MA), and autoregressive moving average (ARMA) models are classic methods of time series analysis. They differ in terms of their autocorrelation and partial autocorrelation functional properties.

A stable normal random sequence can be represented using an ARMA(p, q) model:(6)$$\begin{array}{c} x_{n}=\phi_{1}x_{n-1}+\phi_{2}x_{n-2}+\cdots +\phi_{p}x_{n-p}+\epsilon_{n}+\theta_{1}\epsilon_{n-1}+\\ \theta_{2}\epsilon_{n-2}+\cdots +\theta_{q}\epsilon_{n-q},\epsilon_{i}∼W\left(0,\sigma^{2}\right) \end{array}$$
where $x_{i}$ represents the random variable, $\phi_{i}<1\left(i=1,2\cdots ,p\right)$ are the autoregressive coefficients, $\theta_{i}<1\left(i=1,2\cdots ,q\right)$ are the moving average coefficients, and $\epsilon_{i}$ is a white noise sequence with a mean of 0 and a variance of $\sigma^{2}$.

When $\phi_{i}=0$, the ARMA model simplifies to an MA(q) model of order q:(7)$$x_{n}=\epsilon_{n}+\theta_{1}\epsilon_{n-1}+\theta_{2}\epsilon_{n-2}+\cdots +\theta_{q}\epsilon_{n-q},\epsilon_{i}∼W\left(0,\sigma^{2}\right)$$

When $\theta_{i}=0$, the ARMA model simplifies to an AR(p) model of order p:(8)$$x_{n}=\phi_{1}x_{n-1}+\phi_{2}x_{n-2}+\cdots +\phi_{p}x_{n-p},\epsilon_{i}∼W\left(0,\sigma^{2}\right)$$

In time series model analysis, the first step is to perform a stationarity test on the pressure sensor data. If the data do not meet the stationarity requirement, a first-order or second-order differencing is applied to obtain a stationary signal. Then, the autocorrelation function (ACF) and partial autocorrelation function (PACF) of the sensor data are computed as shown in (9) and (10):(9)$$ACF\left(k\right)=\frac{\frac{1}{N}\sum_{n=1}^{N-k}\left(Y_{n}-\mu_{y}\right)\left(Y_{n+k}-\mu_{y}\right)}{\frac{1}{N}\sum_{n=1}^{N}{\left(Y_{n}-\mu_{y}\right)}^{2}}$$
(10)$$PACF\left(k\right)=\left\{\begin{array}{c} g_{1},k=1\\ \frac{g_{k}\sum_{j=1}^{k-j}g_{k-1,j}g_{k-j}}{1-\sum_{j=1}^{k-j}g_{k-1,j}g_{k}},k=2,3\cdots n \end{array}\right.$$

Here, *k* represents the lag, $g_{k}$ is the sample autocorrelation, $\mu_{y}$ is the sample mean, and $g_{k,k}$ is the sample partial autocorrelation at lag *k*. Time series analysis models are selected based on the autocorrelation function (ACF) and partial autocorrelation function (PACF).

## 3. Airspeed Measurement Device Structure

The MEMS air pressure sensor utilizes the BMP180 sensor manufactured by Bosch Sensortec, Gerlingen, Germany. It comprises a resistive pressure sensor, ADC, control unit, E2PROM, and I2C. The BMP180’s E2PROM stores 11 sets of 16-byte calibration coefficients for temperature and pressure compensation, thereby reducing errors caused by environmental changes, achieving an absolute accuracy of up to 0.03 hPa. It features an eight-PIN ceramic leadless ultra-thin package, with dimensions of only 3.6 mm × 3.8 mm × 0.93 mm, operating voltage ranging from 1.8 V to 3.6 V, and power consumption as low as 3 μA. The BMP180 is characterized by its low cost, high accuracy, small size, and low power consumption. To facilitate airspeed measurement, pressure sensors are installed in the direction of the UAV’s flight to measure total pressure, and similarly, pressure sensors are installed vertically to measure static pressure and temperature. A schematic diagram of the airspeed measurement device structure is shown in Figure 2, and the dimensional parameters are listed in the Table 1. Sensor 1 and Sensor 2 are of the same model, both utilizing the BMP180 pressure sensor from Bosch.

Reference [38] indicated that, when gases flow at high speeds, they form jet streams with complex shockwave structures. Direct contact with the UAV’s body alters the flow field, resulting in sensor measurements that do not represent the flow field at that point, leading to significant errors. Referring to the classic Pitot tube structure, attaching a certain length of tubing to a sensor’s exterior is advantageous in reducing the influence of the rear body of the aircraft on the gas flow field. As mentioned in reference [39], the structure and installation location of an air pressure measurement tube can alter the flow distribution at the static pressure measurement location, resulting in static pressure deviating from the true free flow, known as static pressure error, which is influenced by factors such as the aircraft’s angle of attack and Mach number. Previous studies have demonstrated that extending the air pressure measurement tube can reduce the impact of these variables, as the farther the free flow is from the aircraft’s surface, the smaller the changes in the flow field. Reference [40] embedded a sensor into a soft elastic material to achieve precise airspeed measurement. Inspired by the above literature, attaching a metal tube to the exterior of the air pressure sensor is advantageous in reducing a drone body’s impact on the airflow field, thus obtaining more accurate air pressure values. The structure is shown in Figure 3, and a comparison between the air pressure sensor with and without the metal tube is shown in Figure 4.

## 4. Adaptive Kalman Filter

The KF is commonly used for noise filtering in inertial sensors. The KF is an effective autoregressive filter and an optimal recursive mathematical method that can predict and estimate the current state of a linear dynamic system under a series of incomplete and Gaussian noise measurements. The classic KF relies on accurate knowledge of system noise and measurement noise to achieve optimal estimation. However, in practical applications, obtaining precise statistics for noise is often challenging, leading to a reduction in the accuracy of traditional KFs. Sage–Husa [41] proposed an adaptive Kalman filter, which models measurement noise using an exponentially asymptotic form of the noise covariance matrix, achieving noise-adaptive estimation. Therefore, in this paper, we utilizes the Sage–Husa Kalman filter (SHKF), which can re-estimate and model sensor noise when the model is not sufficiently precise.

### 4.1. Simplified SHKF

The linear system state equation is as follows:(11)$$\left\{\begin{array}{c} X_{k+1}=\phi_{k}X_{k}+w_{k}\\ Z_{k}=H_{k}X_{k}+v_{k} \end{array}\right.$$

The system’s state variables are $X_{k}\in R^{n}$, the system noise is $w_{k}$, the measurement variables are $Z_{k}\in R^{k}$, and the measurement noise is as follows. The system noise and measurement noise are processed into uncorrelated Gaussian random processes with time-invariant mean and time-varying covariance matrices:(12)$$\left\{\begin{array}{c} E\left(w_{k}\right)=\hat{q_{k}},E\left(w_{k}w_{k}^{T}\right)=\hat{Q_{k}}\\ E\left(v_{k}\right)=\hat{r_{k}},E\left(v_{k}v_{k}^{T}\right)=\hat{R_{k}} \end{array}\right.$$

The adaptive SHKF updates $\hat{q_{k}}$, $\hat{Q_{k}}$, $\hat{r_{k}}$, and $\hat{R_{k}}$. Matrix $\hat{Q_{k}}$ represents the variance of the prediction model, while matrix $\hat{R_{k}}$ represents the variance of the measurement model. When used dynamically, as the atmospheric pressure values change rapidly, the filter’s performance deteriorates. To achieve KF adaptation, the filter’s performance is improved by dynamically adjusting matrix $\hat{Q_{k}}$ and matrix $\hat{R_{k}}$ to provide feedback.

Adaptive SHKF calculation flow as shown in Algorithm 1.
**Algorithm 1**: Sage–Husa Kalman filterParameter initialization: ${\hat{X}}_{0}$, ${\hat{P}}_{0}$, ${\hat{r}}_{0}$, ${\hat{R}}_{0}$, ${\hat{q}}_{0}$, ${\hat{Q}}_{0}$, *b*, k=1Main loop:1. Calculate weighted coefficients: $d_{k}=\frac{1-b}{1-b^{k+1}}$2. Predictive error equation for the state: ${\hat{X}}_{k,k-1}=\varphi_{k,k-1}{\hat{X}}_{k-1}+{\hat{q}}_{k-1}$3. Predictive mean square error equation: $P_{k,k-1}=\varphi_{k,k-1}P_{k-1}\varphi_{k,k-1}^{T}+{\hat{Q}}_{k-1}$4. New information sequence equation: ${\hat{r}}_{k}=\left(1-d_{k}\right){\hat{r}}_{k-1}+d_{k}\left(Z_{k}-H_{k}{\hat{X}}_{k,k-1}\right)$5. Measurement noise estimation: ${\hat{R}}_{k}=\left(1-d_{k}\right){\hat{R}}_{k-1}+d_{k}\left[v_{k}v_{k}^{T}-H_{k}P_{k,k-1}H_{k}^{T}\right]$6. Filter gain: $K_{k}=P_{k,k-1}H_{k}^{T}{\left[H_{k}P_{k,k-1}H_{k}^{T}+{\hat{R}}_{k}\right]}^{-1}$7. State estimation: ${\hat{X}}_{k}={\hat{X}}_{k,k-1}+K_{k}v_{k}$8. Estimation mean square error equation: $P_{k}=\left[I-K_{k}H_{k}\right]P_{k,k-1}$9. Model error:$${\hat{q}}_{k}=\left(1-d_{k}\right){\hat{q}}_{k-1}+d_{k}\left({\hat{X}}_{k}-\varphi_{k,k-1}{\hat{X}}_{k-1}\right)$$
$${\hat{Q}}_{k}=\left(1-d_{k}\right){\hat{Q}}_{k-1}+d_{k}\left(K_{k}v_{k}v_{k}^{T}K_{k}^{T}+P_{k}-\varphi_{k,k-1}P_{k-1}\varphi_{k,k-1}^{T}\right)$$

### 4.2. Improved Sage–Husa Adaptive Kalman Filter

#### 4.2.1. Impact of Initial Noise Values

To analyze the impact of measurement noise on the convergence performance of the KF, the following analysis was conducted.

The error of the filter is defined as
(13)$${\bar{X}}_{k}={\hat{X}}_{k}-X_{k}$$

By subtracting ${\hat{X}}_{k}$ from $X_{k+1}$, we obtain
(14)$${\bar{X}}_{k}=\varphi_{k,k-1}{\bar{X}}_{k-1}+{\hat{q}}_{k-1}+K_{k}v_{k}-w_{k-1}$$

Substituting ${\hat{X}}_{k}$ and ${\hat{X}}_{k,k-1}$ into ${\hat{q}}_{k}$, we obtain
(15)$${\hat{q}}_{k}={\hat{q}}_{k-1}+K_{k}d_{k}v_{k}$$

Substituting $Z_{k}$ and ${\hat{X}}_{k,k-1}$ into ${\hat{r}}_{k}$, we obtain
(16)$${\hat{r}}_{k}={\hat{r}}_{k-1}+d_{k}v_{k}$$

Substituting $X_{k+1}$, $Z_{k}$, and ${\hat{X}}_{k,k-1}$ into $v_{k}$, we obtain
(17)$$v_{k}=-H_{k}\varphi_{k,k-1}{\bar{X}}_{k-1}-H_{k}{\hat{q}}_{k-1}-r_{k-1}+H_{k}w_{k-1}+v_{k}$$

Substituting (17) into (14) through (16), we obtain:   
(18)$$\begin{array}{c} {\bar{X}}_{k}=\varphi_{k,k-1}\left(1-K_{k}H_{k}\right){\bar{X}}_{k-1}+\left(1-K_{k}H_{k}\right){\hat{q}}_{k-1}-\\ K_{k}{\hat{r}}_{k-1}+\left(K_{k}H_{k}-1\right)w_{k-1}+K_{k}v_{k} \end{array}$$
(19)$$\begin{array}{c} {\hat{q}}_{k}=-K_{k}d_{k}H_{k}\varphi_{k,k-1}{\bar{X}}_{k-1}+\left(1-K_{k}d_{k}H_{k}\right){\hat{q}}_{k-1}-\\ K_{k}d_{k}r_{k-1}+K_{k}d_{k}H_{k}w_{k-1}+K_{k}d_{k}v_{k} \end{array}$$
(20)$$\begin{array}{c} {\hat{r}}_{k}=-d_{k}H_{k}\varphi_{k,k-1}{\bar{X}}_{k-1}-d_{k}H_{k}{\hat{q}}_{k-1}+\left(1-d_{k}\right){\hat{r}}_{k-1}+d_{k}H_{k}w_{k-1}+d_{k}v_{k} \end{array}$$

Rearranging (18)–(20) into the following form results in a linear system state equation composed of $X_{k}$, ${\hat{q}}_{k}$, and ${\hat{r}}_{k}$:(21)$$\begin{array}{c} \left[\begin{array}{c} {\bar{X}}_{k}\\ {\hat{q}}_{k}\\ {\hat{r}}_{k} \end{array}\right]=\left[\begin{array}{ccc} \varphi_{k,k-1}\left(1-K_{k}H_{k}\right) & \left(1-K_{k}H_{k}\right) & -K_{k}\\ -K_{k}d_{k}H_{k}\varphi_{k,k-1} & \left(1-K_{k}d_{k}H_{k}\right) & -K_{k}d_{k}\\ -d_{k}H_{k}\varphi_{k,k-1} & -d_{k}H_{k} & \left(1-d_{k}\right) \end{array}\right]\left[\begin{array}{c} {\bar{X}}_{k-1}\\ {\hat{q}}_{k-1}\\ {\hat{r}}_{k-1} \end{array}\right]\\ +\left[\begin{array}{cc} \begin{array}{c} \left(K_{k}H_{k}-1\right)\\ K_{k}d_{k}H\\ d_{k}H_{k} \end{array} & \begin{array}{c} K_{k}\\ K_{k}d_{k}\\ d_{k} \end{array} \end{array}\right]\left[\begin{array}{c} w_{k-1}\\ v_{k} \end{array}\right] \end{array}$$

Equation (Equation 17) can be rearranged as
(22)$$v_{k}=\left[\begin{array}{ccc} -H_{k}\varphi_{k,k-1} & -H_{k} & -1 \end{array}\right]\left[\begin{array}{c} {\bar{X}}_{k-1}\\ {\hat{q}}_{k-1}\\ {\hat{r}}_{k-1} \end{array}\right]+\left[\begin{array}{cc} H_{k} & 1 \end{array}\right]\left[\begin{array}{c} w_{k-1}\\ v_{k} \end{array}\right]$$

Where the state transition matrix is
(23)$$A=\left[\begin{array}{ccc} \varphi_{k,k-1}\left(1-K_{k}H_{k}\right) & \left(1-K_{k}H_{k}\right) & -K_{k}\\ -K_{k}d_{k}H_{k}\varphi_{k,k-1} & \left(1-K_{k}d_{k}H_{k}\right) & -K_{k}d_{k}\\ -d_{k}H_{k}\varphi_{k,k-1} & -d_{k}H_{k} & \left(1-d_{k}\right) \end{array}\right]$$

Calculating the eigenvalues of the state matrix *A*, we obtain
(24)$$\left|\lambda I-A\right|=\left(\lambda -1\right)\left|\begin{array}{ccc} \lambda -\varphi_{k,k-1}\left(1-K_{k}H_{k}\right) & \left(K_{k}H_{k}-1\right) & K_{k}\\ 0 & 1 & -K_{k}\\ d_{k}H_{k}\varphi_{k,k-1} & d_{k}H_{k} & \lambda -1+d_{k} \end{array}\right|$$

A state transition matrix *A* always has an eigenvalue of 1. When the eigenvalues of the state transition matrix of a linear system are equal to 1, the system is in a critically stable state. The state prediction error ${\hat{q}}_{k}$, measurement error ${\hat{r}}_{k}$, and system error ${\bar{X}}_{k}$, together, constitute a stable system, meaning that errors in ${\hat{q}}_{k}$ and ${\hat{r}}_{k}$ can lead to an increase in the system error ${\bar{X}}_{k}$. When the initial values ${\hat{q}}_{0}$ and ${\hat{r}}_{0}$ and the values of ${\hat{q}}_{k}$ and ${\hat{r}}_{k}$ have significant errors, the impact of initial value errors ${\bar{X}}_{k}$ can result in a substantial and persistent system error that cannot be avoided.

In this paper, the values of ${\hat{q}}_{0}$ and ${\hat{r}}_{0}$ are determined using the time series model of the sensor. The covariance of the original signal is set as the measurement covariance ${\hat{r}}_{0}$, and the covariance of the residual signal after the AR(3) model is set as the system covariance ${\hat{q}}_{0}$.

#### 4.2.2. Impact of Data Outliers

When using a KF for filtering, sensor noise is typically treated as white noise or Gaussian noise. However, during the measurement process, outliers can appear in the measurements due to issues with the sensor itself or external factors, leading to bias or divergence in the filter. When the outliers are relatively small compared to the sensor measurements, the values of ${\hat{r}}_{k}$ and $v_{k}$ are also small, resulting in a correspondingly small value for $v_{k}v_{k}^{T}$. However, when the outliers are significantly larger than the sensor measurements, the value of $P_{k,k-1}$ in the one-step predictive mean square error equation becomes larger. In both scenarios, this can lead to a negative value for ${\hat{R}}_{k}$, causing the filter to behave abnormally.

To prevent the filter from being adversely affected and diverging due to sensor outliers, this paper employs a strategy of setting the sensor measurement noise within a range $\left[{\hat{R}}_{k\min},{\hat{R}}_{k\max}\right]$ and correcting noise values $R_{k}$ that fall outside this range. This approach helps prevent filter divergence caused by large fluctuations in measurement noise.

The correction process for $R_{k}$ is represented by (25) shown below, where $\beta_{k}$ is the measurement noise correction coefficient and $0<\beta_{k}<1$:(25)R^k=R^kmin,  R^k<R^kminβkR^k,  R^k>R^kmax  R^k,  R^kmin<R^k<R^kmax

Combining the AR method with the improved AR-SHAKF forms the flowchart for the improved AR-SHAKF method, as shown in the Algorithm 2.
**Algorithm 2**: Improved AR-SHAKF filterInput: Sensor static data, sensor measurement data1. Sensor static data stationary test2. Data preprocessing3. AR model order selection: examine ACF, PACF, and AIC4. Parameter estimation: Yule–Walker approach5. Determine ${\hat{q}}_{0}$ and ${\hat{r}}_{0}$6. Given initial value of ${\hat{X}}_{0}$, ${\hat{P}}_{0}$, ${\hat{R}}_{0}$, ${\hat{Q}}_{0}$, and b7. Calculated weighted coefficient: $d_{k}$8. Predicted state: ${\hat{X}}_{k,k-1}$,Predicted mean square error equation: $P_{k,k-1}$9. New information sequence equation: ${\hat{r}}_{k}$,$v_{k}$10. Measurement noise estimation: ${\hat{R}}_{k}$Measurement noise correction:R^k=R^kmin,  R^k<R^kminβkR^k,  R^k>R^kmax  R^k,  R^kmin<R^k<R^kmax11. Filtering gain: $K_{k}$State estimation algorithm: ${\hat{X}}_{k}$Mean square error of estimation: $P_{k}$12. Model error: ${\hat{q}}_{k}$, ${\hat{Q}}_{k}$Output: filter output ${\hat{X}}_{k}$

#### 4.2.3. Algorithm Complexity Analysis

To compare the complexity of the KF and the improved AR-SHAKF filtering algorithm proposed in this paper, we used the Big-O notation introduced by Juris Hartmanis and Richard E. Stearns to calculate complexity. In Big-O notation, O represents the order of magnitude, and O(f(n)) indicates how the complexity of the algorithm increases with the problem size n.

The complexity of the KF algorithm is mainly determined by the matrix operations in the prediction and update steps. The complexity of the prediction step is $O\left(n^{3}\right)$, and the complexity of the update step is $O\left(n^{3}+mn^{2}+m^{3}\right)$. Typically, the state dimension n is greater than the observation dimension m, so $n^{3}$ dominates the algorithm’s complexity. Therefore, the complexity of the KF algorithm is $O\left(n^{3}\right)$.

The complexity of the improved AR-SHAKF filtering algorithm proposed in this paper is divided into two parts: the AR model and the adaptive Kalman filter. The complexity of the autoregressive (AR) model is $O\left(n^{2}\right)$, and the complexity of the adaptive KF algorithm is $O\left(n^{3}+mn^{2}+m^{3}\right)$. Therefore, the overall complexity of the improved AR-SHAKF algorithm is $O\left(n^{3}+mn^{2}+m^{3}+n^{2}\right)$. Similarly, since the complexity is dominated by $n^{3}$, the complexity of the improved AR-SHAKF algorithm is also $O\left(n^{3}\right)$. Thus, it can be seen that the complexities of the KF and improved AR-SHAKF algorithms are similar.

The improved AR-SHAKF algorithm involves extensive matrix computations, requiring the microprocessor to have efficient computation units, support for single and double precision, and capabilities for parallel operations such as floating-point calculations. Additionally, it necessitates parallel processing capabilities like multi-core architecture and hardware threading, as well as sufficient memory bandwidth and capacity, including high-speed memory access, large memory capacity, and low memory latency.

## 5. Experimental Analysis

### 5.1. Allan Variance Analysis Method

To analyze the filtering effectiveness of the AR-SHAKF method proposed in this paper, the Allan variance analysis method was utilized to compare the data collected during static experiments with the filtered data. The Allan variance method does not require any additional transformations and allows for the characterization of noise in sensor time-domain signals as a function of averaging time. It also aids in determining the type of noise [42,43]. When calculating the Allan variance, the sensor signal is divided into multiple groups of fixed sample lengths in the time domain, and the variance is computed within groups of the same size. The Allan variance for *N* sampled signals is given by
(26)$$\begin{array}{c} \sigma^{2}\left(\tau \right)=\frac{1}{2\left(N-2n\right)}\sum_{k=1}^{N-2n}{\left[{\bar{\Omega }}_{k+1}\left(\tau \right)-{\bar{\Omega }}_{k}\left(\tau \right)\right]}^{2} \end{array}$$
where *N* is the total number of sensor data points, and all the data are divided into several groups. ${\bar{\Omega }}_{k}\left(\tau \right)$ represents the average value of the data in the *k*-th group.

### 5.2. Static Experiments

Sensor 1 and Sensor 2, designated for measuring total pressure and static pressure, respectively, were statically positioned indoors at a temperature of 25 °C for one hour before data collection commenced. Efforts were made to maintain a relatively quiet environment during the data acquisition process. The static experiment is depicted in Figure 5. The sampling period for sensor data was set at 10 ms, with an output data rate of 10 Hz. Data were continuously collected for half an hour. Both sensors were configured with identical sampling periods and connected to the control chip. The pressure and temperature data obtained from the sensors were transmitted to the control chip via the I2C interface. Subsequently, the control chip transmitted the data to the computer through a serial port. The acquired data are illustrated in Figure 6.

A segmented fitting of the Allan variance for the sensors resulted in the Allan variances for both sensors, as shown in Figure 7. The red, purple, and orange lines in the figure represent the reference lines for −1, 0, and 1, respectively.

According to Figure 7, during the initial period, the slopes of the Allan variance curves for both Sensor 1 and Sensor 2 are approximately $-\frac{1}{2}$. Random walk and white noise are the primary sources of error for the sensors during this short clustering time. Subsequently, the sensors briefly exhibit bias instability. In the time that follows, the slope of the sensor curves changes to $\frac{1}{2}$, and the noise continues to exhibit characteristics of a random walk.

Based on the static laboratory data, time series models for the atmospheric pressure sensors were determined. The autocorrelation functions (ACFs) and partial autocorrelation functions (PACFs) for both sensors were calculated and plotted, as shown in Figure 8. It can be observed that the ACFs for Sensors 1 and 2 exhibit a trailing characteristic, while the PACFs display a truncating characteristic. This suggests the selection of an autoregressive model AR(p) to represent the sensor state equations.

The AR(p) requires model order determination and parameter estimation. The Akaike information criterion (AIC) was used for model order determination. The AIC is defined as shown in (27), where *N* represents the sample size, *p* represents the order of model parameters, *L* represents the likelihood function, and the goal is to select the best value of p that minimizes the AIC. Once the order p is determined, the Yule–Walker method is used to calculate the model parameters and the corresponding AIC values for Sensors 1 and 2, as shown in Table 2 and Table 3.
(27)$$AIC\left(p\right)=2p-2\ln\left(L\right)$$

It can be observed that, as the order of the AR model increases, the AIC value significantly decreases, indicating that the model gets closer to the true signal. However, with the increase in order, the computational demands on the system also increase. Considering the requirements for low-cost unmanned aerial vehicles, including simple hardware and computational speed, an AR(3) model was chosen to establish the sensor system state equation. Since the measurement signal from the sensor is not zero-mean, a constant term ‘*c*’ is introduced as a state variable in the state matrix. The state vector is then given as follows:(28)$$X_{k}=\left[\begin{array}{c} x_{k}\\ x_{k-1}\\ x_{k-2}\\ c \end{array}\right]$$

The AR(p) model can be represented as
(29)$$x_{k}=\varphi_{1}x_{1}+\varphi_{2}x_{2}+\cdots +\varphi_{p}x_{k}+c+a_{k}$$

Using the AR(3) model, $x_{k}$ becomes
(30)$$\begin{array}{c} \left[\begin{array}{c} x_{k}\\ x_{k-1}\\ x_{k-2}\\ c \end{array}\right]=\left[\begin{array}{cccc} \varphi_{1} & \varphi_{2} & \varphi_{3} & 1\\ 1 & 0 & 0 & 0\\ 0 & 1 & 0 & 0\\ 0 & 0 & 0 & 1 \end{array}\right]\left[\begin{array}{c} x_{k-1}\\ x_{k-2}\\ x_{k-3}\\ c \end{array}\right]+\left[\begin{array}{cccc} 1 & 0 & 0 & 0\\ 0 & 0 & 0 & 0\\ 0 & 0 & 0 & 0\\ 0 & 0 & 0 & 0 \end{array}\right]\left[\begin{array}{c} a_{k}\\ 0\\ 0\\ 0 \end{array}\right] \end{array}$$

$Z_{k}$ is:(31)$$Z_{k}=\left[\begin{array}{cccc} 1 & 0 & 0 & 0 \end{array}\right]\left[\begin{array}{c} x_{k-1}\\ x_{k-2}\\ x_{k-3}\\ c \end{array}\right]+v_{k}$$

By using the improved AR-SHAKF and the standard KF to filter the static sensor data, the filtered data are obtained and presented in Figure 9.

The performance of the algorithms was compared using the maximum error (ME), mean absolute error (MAE), and root mean square error (RMSE) values, as defined in (32)–(34). ME represents the maximum error value; MAE is the average of absolute errors and reflects the actual error before and after filtering; RMSE is the square root of the arithmetic mean of the squared errors before and after filtering. Smaller values for these metrics indicate better filtering performance of the model.
(32)$$\mathrm{ME}=\max\left|x\left(i\right)-\bar{x}\left(i\right)\right|$$
(33)$$\mathrm{MAE}=\frac{1}{N}\sum_{i=1}^{N}\left|x\left(i\right)-\bar{x}\left(i\right)\right|$$
(34)$$\mathrm{RMSE}=\sqrt{\frac{1}{N}\sum_{i=1}^{N}{\left(x\left(i\right)-\bar{x}\left(i\right)\right)}^{2}}$$
where $\bar{x}\left(i\right)$ is the signal mean. $x\left(i\right)$ is the actual signal value. *N* is the signal length.

Based on the data in Table 4, it can be observed that both the KF and the improved AR-SHAKF effectively filter the sensor data. The improved AR-SHAKF has smaller ME, MAE, and RMSE values compared to the KF, indicating that the filtering performance is superior, and the signal-to-noise ratio is improved.

The Allan variance analysis of the filtered sensor static data resulted in a logarithmic time plot, as shown in Figure 10. Using the least squares method, the noise coefficients for the sensor before and after filtering were fitted, and the results are presented in Table 5. It can be observed that, in the initial short time of the operation, the sensor no longer contains random walk white noise. After a longer operating time, although the sensor contains random walk noise, the noise values significantly decrease. This indicates that the improved AR-SHAKF filtering method effectively reduces the noise in MEMS atmospheric pressure sensors, which is crucial for airspeed calculations in unmanned aerial vehicles.

### 5.3. Dynamic Experiments

To validate the effectiveness of the mechanical and algorithmic filtering proposed in this paper, an airspeed measurement device was designed and fabricated. Before conducting the wind tunnel tests, the wind speed and duration of the wind tunnel were set using a pressure control valve with closed-loop control. This value was then plotted over time to represent the true airspeed measurement. The wind tunnel was set to a specific airspeed as the true airspeed, the airspeed calculated from measurements was considered the measured airspeed, and the airspeed calculated after KF and improved AR-SHAKF filtering was used as the KF-calculated airspeed and improved AR-SHAKF-calculated airspeed, respectively.

The experiments were divided into two groups. In the first group of experiments, the airspeed measurement device had metal pipes installed. Measurements of airspeed were calculated, and airspeed was separately calculated using the KF and improved AR-SHAKF algorithms. This analysis aimed to assess the filtering effectiveness of the improved AR-SHAKF algorithm. In the second group of experiments, the airspeed measurement device did not have metal pipes installed around the sensors. This analysis was conducted to evaluate the effectiveness of the mechanical filtering structure designed in this paper.

#### 5.3.1. Experiment 1: Verification of the AR-SHAKF Algorithm Filtering Effect

The airspeed measurement device was installed as shown in Figure 11. The wind tunnel airflow was directed to the right, with the total pressure measurement channel facing the airflow direction to ensure that was is aligned with the center of the wind tunnel. The static pressure measurement channel was perpendicular to the airflow direction, and metal tubes were added externally to the sensors. The wind tunnel wind speed was set in multiple stages, starting from low to high speed, with each speed level maintained for a certain duration. The measurements for total pressure by Sensor 1, static pressure by Sensor 2, and temperature data are shown in Figure 12.

The total pressure shows a step-like increase, while the static pressure decreases slightly. The temperature variation is also significant, with a sharp drop during blowing and a gradual recovery to room temperature when there is no wind. From these measurements, it can be inferred that total pressure is most sensitive to changes in wind speed, static pressure is almost unaffected by wind speed, and temperature can also reflect the trend of wind speed changes.

The reference [44] utilizes a Kalman filter for sensor noise reduction. In this paper, we employed the Allan variance method to analyze the primary sensor noise [45], identifying it as random walk noise. While the Kalman filter models sensor noise as a fixed value and do not consider the impact of noise variations on the filter, they can result in significant errors between measured airspeed and actual airspeed values. Additionally, random errors cannot be eliminated through calibration and require signal modeling before filtering. Therefore, we adopted a time-regression adaptive model to make the filtered airspeed values closer to the true values. The effectiveness comparison between the proposed improved AR-SHAKF filter and the Kalman filter is depicted in Figure 13, while the comparison of ME, MAE, and RMSE for both filtering methods is presented in Table 6.

Both the KF and the improved AR-SHAKF effectively reduce sensor noise. For the dynamically fluctuating dynamic pressure signal, the KF reduces the ME, MAE, and RMSE by 1.42%, 0.21%, and 0.12%, respectively. The improved AR-SHAKF, on the other hand, reduces these errors by 2.73%, 0.69%, and 0.67%, respectively. The improved AR-SHAKF filter exhibits better filtering performance than the KF, with significantly reduced ME, MAE, and RMSE values in the filtered signals.

To analyze the advantages of the improved AR-SHAKF method proposed in this paper, we compared the airspeed values calculated using this method with those obtained from other methods. In reference [46], a first-order lag filter is employed to optimize the pressure values measured by the sensor. The filtering equation is as follows:(35)$$Y_{n}=\left(1-k\right)X_{n}+kY_{n-1}$$

In the equation, $Y_{n}$ represents the current filtering output value, $X_{n}$ represents the current readout data, $Y_{n-1}$ represents the previous filtering output value, and *k* is a filtering coefficient ranging from 0 to 1.

In reference [47], based on the recursive averaging filtering algorithm, N data points are allocated with weighting coefficients to form a weighted recursive averaging filtering (WRAF) algorithm, as shown below:(36)$$Z_{n}=k_{1}X_{n}+k_{1}X_{n-1}+k_{3}X_{n-2}$$

In the equation, $Z_{n}$ represents the current filtering output value, $X_{n}$, $X_{n-1}$, and $X_{n-2}$ represent the sensor measurement values at times n, n − 1, and n − 2, respectively. $k_{1}$, $k_{2}$, and $k_{3}$ are the weighting coefficients satisfying $k_{1}+k_{2}+k_{3}=1$.

The airspeed measurements were processed for noise reduction using the improved AR-SHAKF filter, KF, FOLF, and WRAF. The filtered data were then compared with the true airspeed values, as depicted in Figure 14.

In (32)–(34), the values for ME, MAE, and RMSE may not adequately reflect the differences between the signal and the true value. Based on this, further modifications were made to (35)–(37) as formulas for assessing the effectiveness of the filtering method, and the results are shown in Table 7.
(37)$${\mathrm{ME}}^{\prime }=\max\left|x_{1}\left(i\right)-x_{v}\left(i\right)\right|$$
(38)$${\mathrm{MAE}}^{\prime }=\frac{1}{N}\sum_{i=1}^{N}\left|x_{1}\left(i\right)-x_{v}\left(i\right)\right|$$
(39)$${\mathrm{RMSE}}^{\prime }=\sqrt{\frac{1}{N}\sum_{i=1}^{N}{\left(x_{1}\left(i\right)-x_{v}\left(i\right)\right)}^{2}}$$

In the equation, $x_{1}\left(i\right)$ is the airspeed value from Experiment 1, and $x_{v}\left(i\right)$ represents the true airspeed. The error rate for ME’ is defined as shown in the equation, and the error rates for MAE’ and RMSE’ are calculated accordingly.
(40)$$\varepsilon_{ME\left(og-KF\right)}=\frac{\left|M{E^{\prime }}_{og}-M{E^{\prime }}_{KF}\right|}{M{E^{\prime }}_{og}}$$

Comparing the ME, MAE, and RMSE of the filtered signals with the true signal values reveals that the error is smallest for the signal filtered using the improved AR-SHAKF filter, followed by the KF. The FOLF and WRAF exhibit the largest errors. The improved AR-SHAKF filter demonstrates the greatest reduction in airspeed value error after noise reduction, resulting in airspeed values that are closer to the true values. Among the four filters, the filtering effect of the improved AR-SHAKF filter is optimal, providing the best noise removal performance.

#### 5.3.2. Experiment 2: Mechanical Filtering Effect Validation

To analyze the filtering effectiveness of the mechanical structure designed in this paper, total pressure and static pressure measurements were directly taken from the wind tunnel using an airspeed measurement device without metal tubes. The installation method and position were the same as in Experiment 1, as shown in Figure 15. The measured total pressure, static pressure, and temperature are displayed in Figure 16.

Compared with Experiment 1, it can be observed that, when there are no metal tubes outside the sensors, the measured total pressure and static pressure signals exhibit significant fluctuations, and there is more sensor noise. This is because, without metal tubes, the sensors are influenced by airflow from multiple directions, leading to significant interference in the measurement results of air pressure and temperature. This interference has a significant impact on the airspeed calculation accuracy. The mechanical filtering device designed in this paper serves as a physical filter to mitigate these effects.

According to the measured total pressure, static pressure, and temperature data, the comparison between the calculated airspeed and set airspeed is shown in Figure 17. The formulas for calculating the ME′′, MAE′′, and RMSE′′ of airspeed are shown in (39)–(41).
(41)$${\mathrm{ME}}^{\prime \prime }=\max\left|x_{2}\left(i\right)-x_{v}\left(i\right)\right|$$
(42)$${\mathrm{MAE}}^{\prime \prime }=\frac{1}{N}\sum_{i=1}^{N}\left|x_{2}\left(i\right)-x_{v}\left(i\right)\right|$$
(43)$${\mathrm{RMSE}}^{\prime \prime }=\sqrt{\frac{1}{N}\sum_{i=1}^{N}{\left(x_{2}\left(i\right)-x_{v}\left(i\right)\right)}^{2}}$$

In the equations, $x_{2}\left(i\right)$ represents the airspeed value from Experiment 2. To compare the differences in calculating airspeed with and without metal tubes, the error rate calculation formulas for Experiment 1 and Experiment 2 are defined as follows: (44)$$\varepsilon_{ME}=\frac{ME^{\prime \prime }-ME^{\prime }}{ME^{\prime }}$$
(45)$$\varepsilon_{MAE}=\frac{MAE^{\prime \prime }-MAE^{\prime }}{MAE^{\prime }}$$
(46)$$\varepsilon_{RMSE}=\frac{RMSE^{\prime \prime }-RMSE^{\prime }}{RMSE^{\prime }}$$

Combining Figure 17 and Table 8, it can be seen that the airspeed measurement device without metal tubes exhibits significant measurement noise, resulting in a low signal-to-noise ratio and airspeed calculation results deviating from the actual airspeed. Through experimental verification, it is evident that the mechanical filtering structure designed in this paper can act as a physical filter, converging turbulent airflow into a consistent direction and reducing noise caused by airflow from multiple directions. This device holds significant importance for platforms that require high airspeed calculation accuracy, such as unmanned UAVs.

## 6. Summary

This paper designed a compact airspeed measurement device for small UAVs. Given the small size and low cost of these UAVs, a data processing method combining mechanical filtering and an improved AR-SHAKF algorithm was proposed. The mechanical filtering reduces the impact of the UAV’s body on the airflow by adding a metal tube. The improved AR-SHAKF algorithm establishes a state equation using an autoregressive model based on static sensor data, determines initial noise values, and restricts the measurement noise matrix range to achieve adaptive noise estimation, preventing filter divergence. The algorithm showed significantly lower ME, MAE, and RMSE compared to KF, resulting in more accurate airspeed calculations.

A low-complexity airspeed measurement device based on MEMS pressure sensors was designed and manufactured. Wind tunnel tests analyzed the impact of both filtering methods on the airspeed measurement device’s calculations. The results indicate that both the designed airspeed measurement device and the physical and algorithmic filtering methods effectively calculate airspeed with high accuracy and strong anti-interference characteristics. However, the proposed methods require high hardware computational resources, and further research will be conducted to address this issue.

## Figures and Tables

**Figure 1 sensors-24-05853-f001:**
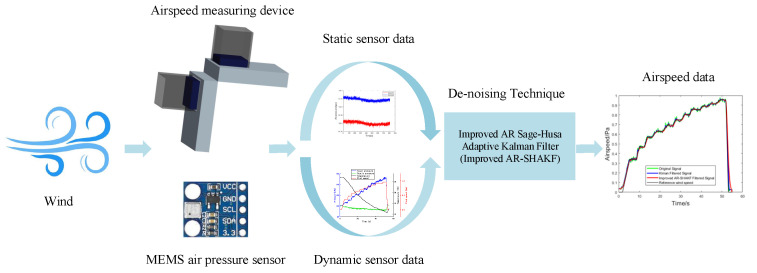
The airspeed measurement flowchart.

**Figure 2 sensors-24-05853-f002:**
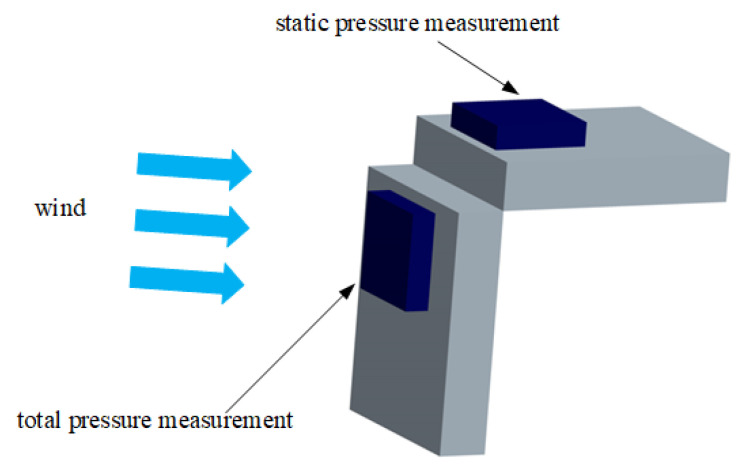
Schematic diagram of airspeed measurement device structure.

**Figure 3 sensors-24-05853-f003:**
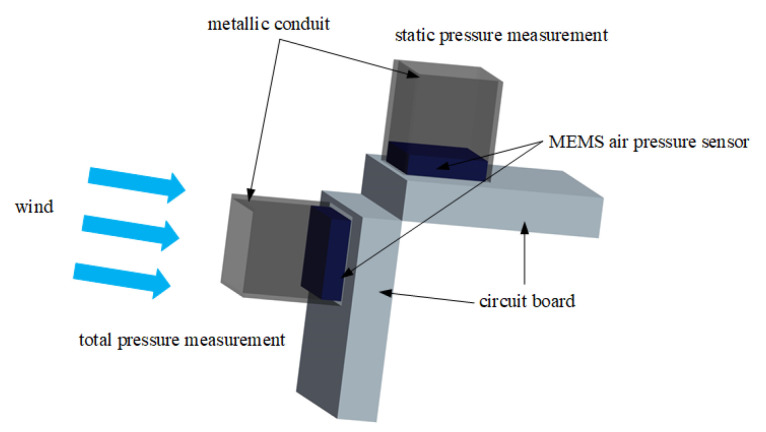
Schematic diagram of improved structure for airspeed measurement device.

**Figure 4 sensors-24-05853-f004:**
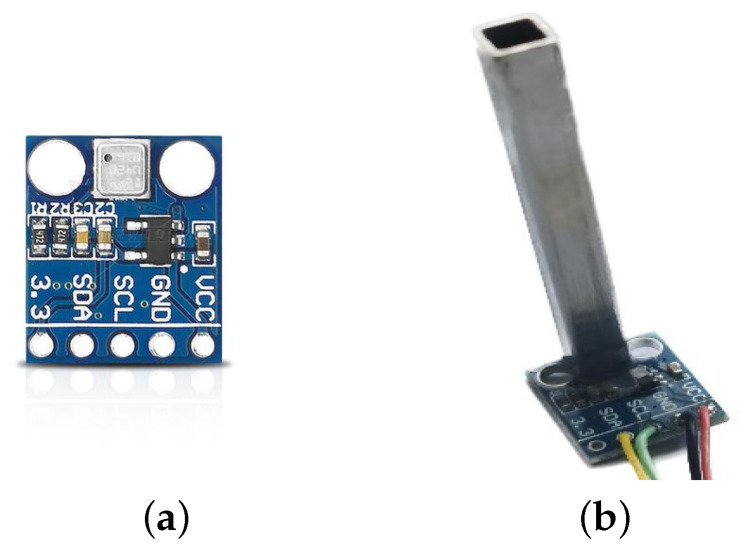
Comparison diagram of air pressure sensor retrofit with metal tube. (**a**) Air pressure sensor; (**b**) Retrofitting metal tube air pressure sensor.

**Figure 5 sensors-24-05853-f005:**
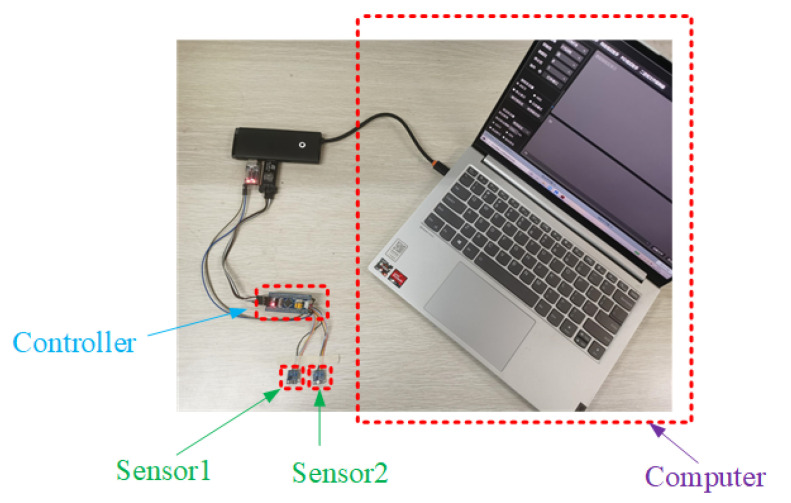
Sensor static experiment setup.

**Figure 6 sensors-24-05853-f006:**
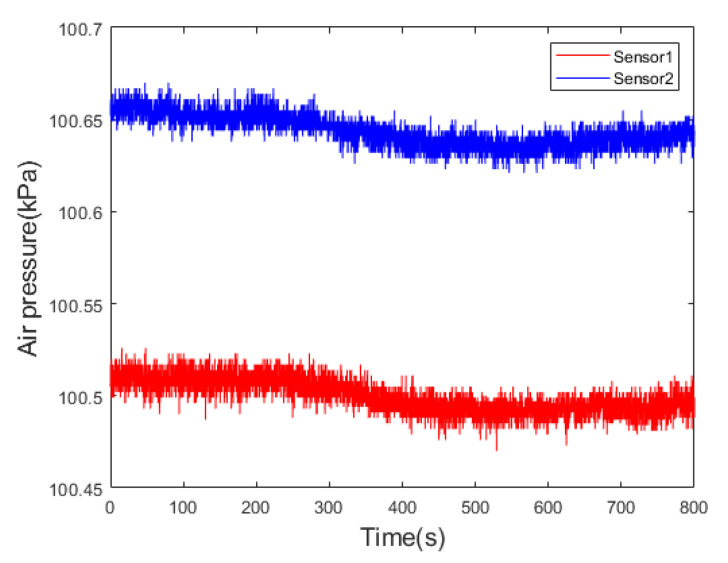
Sensor static measurement data.

**Figure 7 sensors-24-05853-f007:**
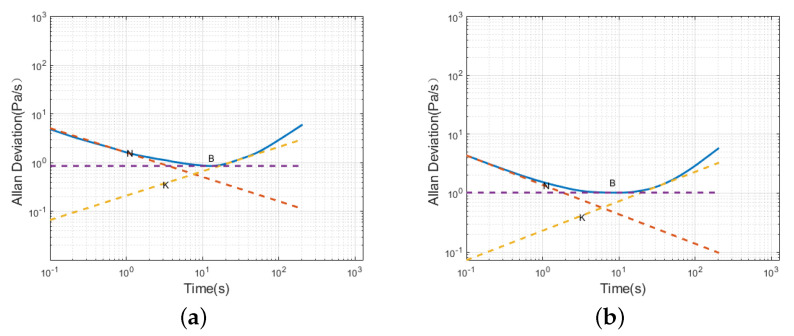
Sensor Allan variance plot; (**a**) Sensor 1; (**b**) Sensor 2.

**Figure 8 sensors-24-05853-f008:**
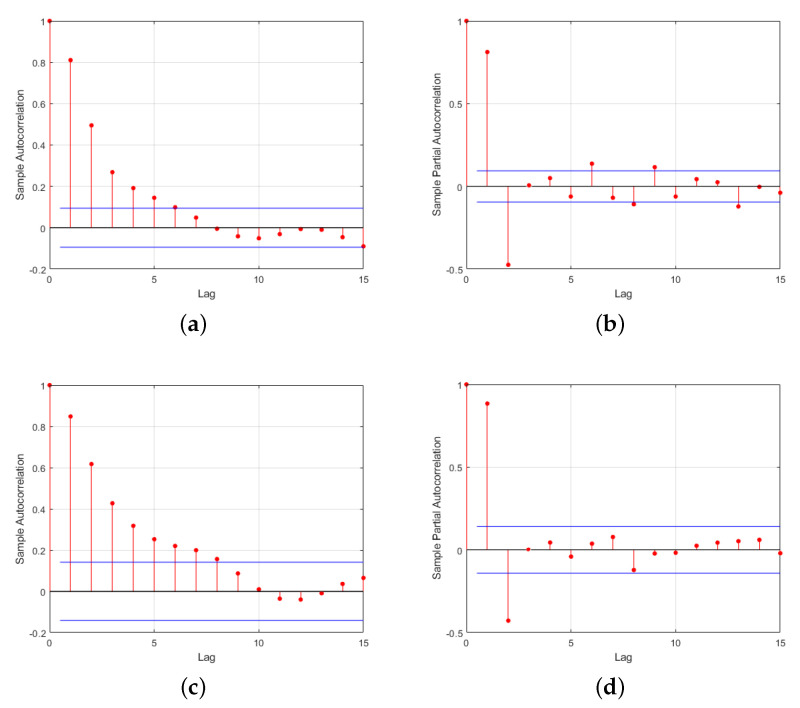
ACF and PACF plots for Sensor 1 and Sensor 2; (**a**) ACF plot for Sensor 1; (**b**) PACF plot for Sensor 1; (**c**) ACF plot for Sensor 2; (**d**) PACF plot for Sensor 2.

**Figure 9 sensors-24-05853-f009:**
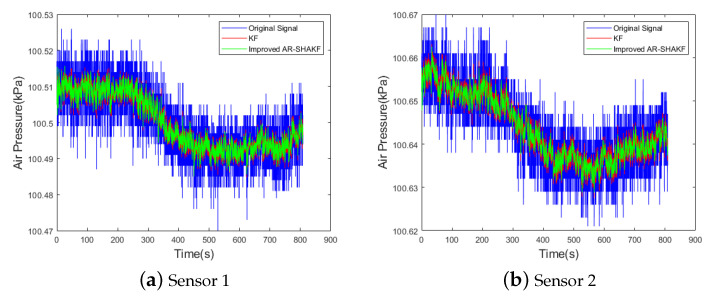
Comparison of static data filtering.

**Figure 10 sensors-24-05853-f010:**
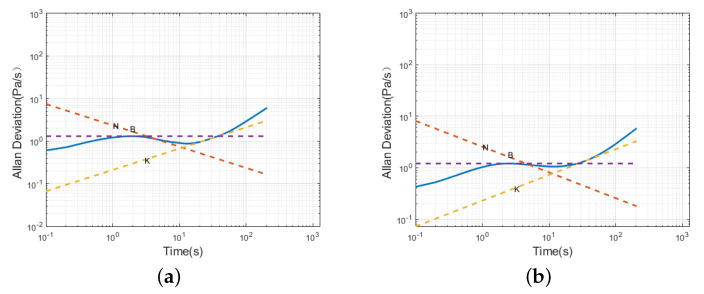
Allan variance plot of filtered static sensor data; (**a**) Sensor 1; (**b**) Sensor 2.

**Figure 11 sensors-24-05853-f011:**
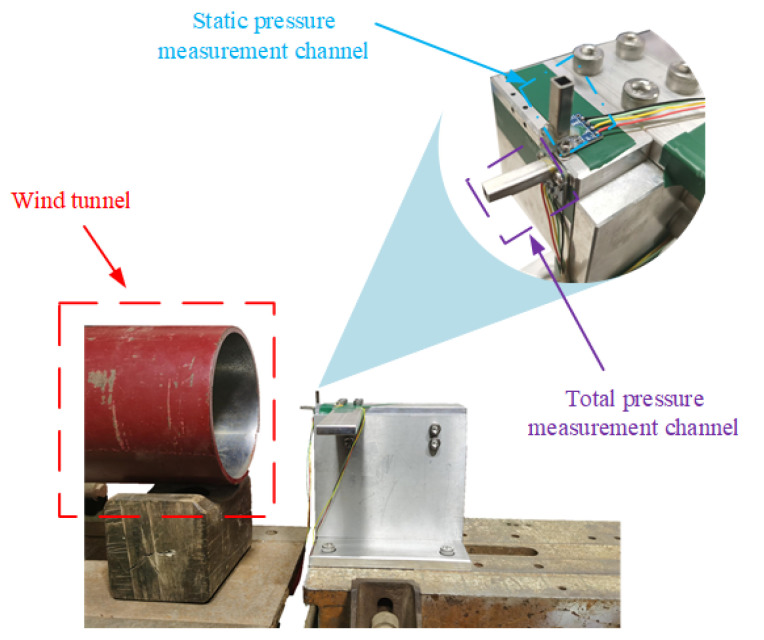
Wind tunnel test diagram of airspeed measurement device.

**Figure 12 sensors-24-05853-f012:**
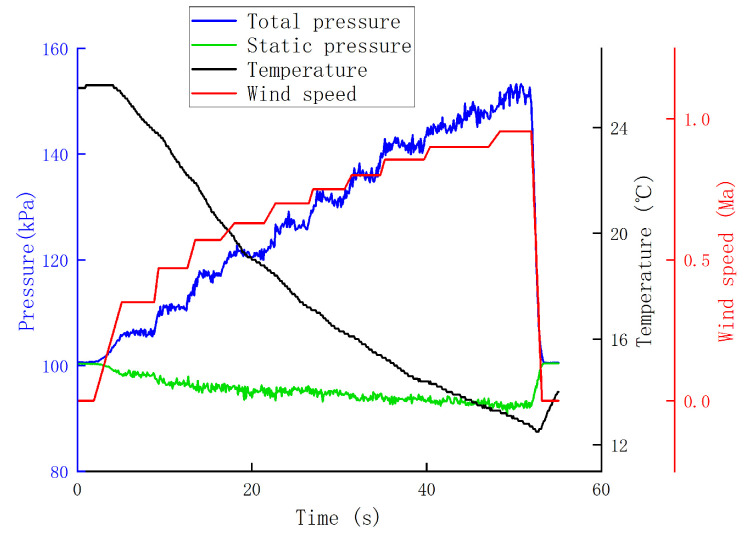
Experiment 1 data.

**Figure 13 sensors-24-05853-f013:**
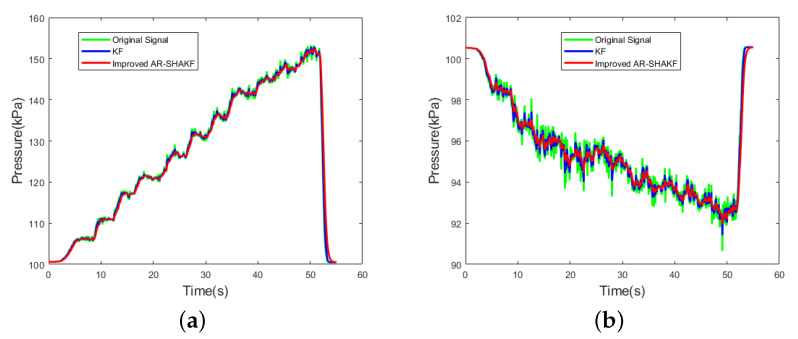
Comparison of sensor signal filtering; (**a**) Total pressure signal; (**b**) Static pressure signal.

**Figure 14 sensors-24-05853-f014:**
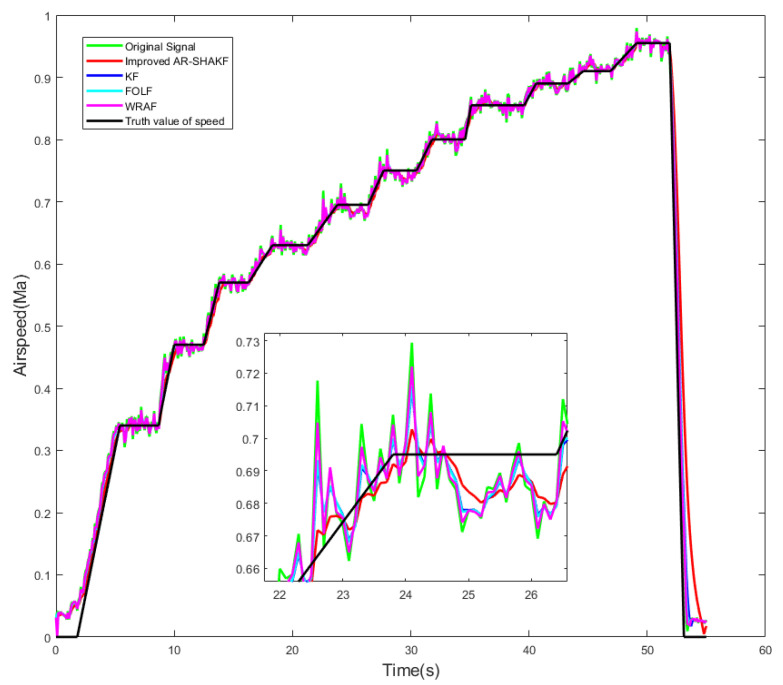
Comparison of calculated airspeed results with true airspeed values.

**Figure 15 sensors-24-05853-f015:**
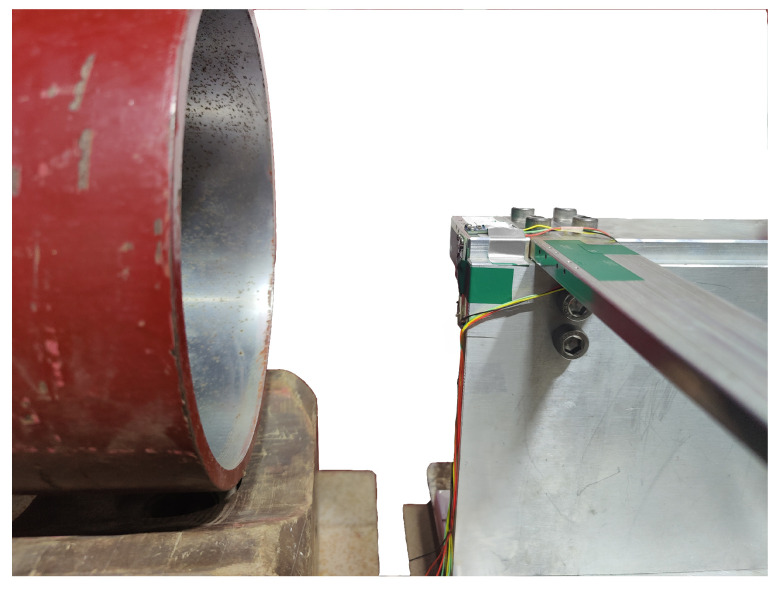
Wind tunnel test diagram of airspeed measurement device without metal tubes.

**Figure 16 sensors-24-05853-f016:**
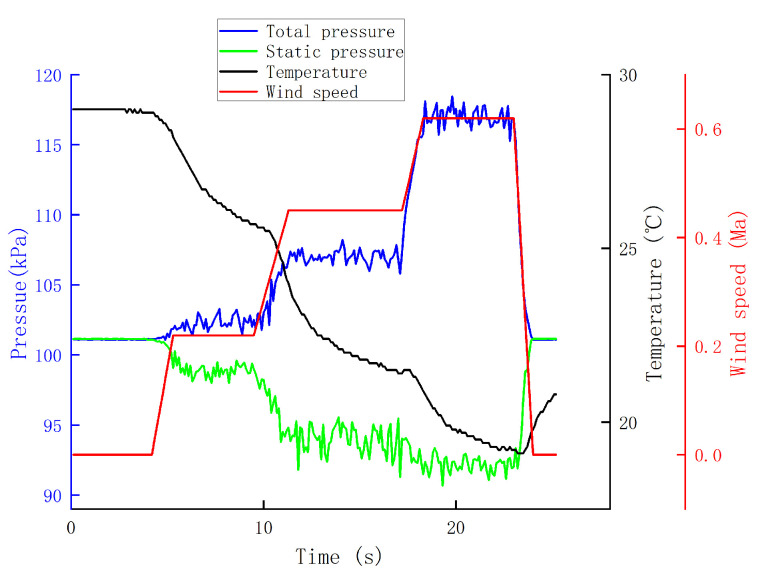
Experiment 2 data.

**Figure 17 sensors-24-05853-f017:**
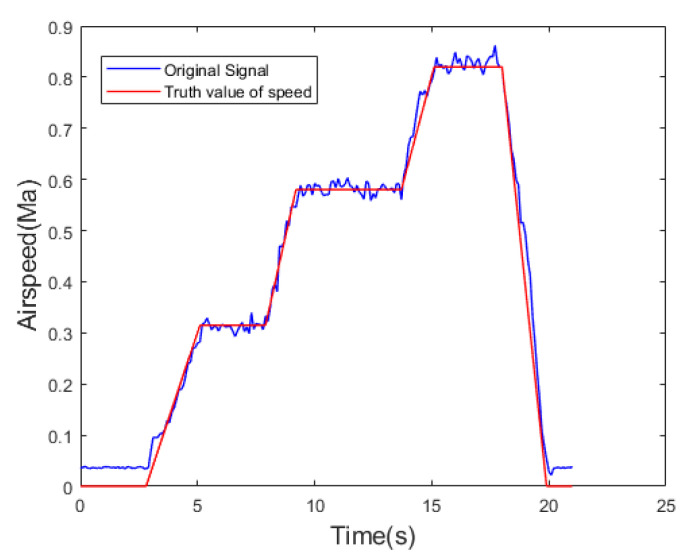
Comparison of airspeed calculation results with true airspeed.

**Table 1 sensors-24-05853-t001:** Airspeed measurement device parameters.

	Bmp180	Circuit Board	Metallic Conduit
Size (mm)	3.6 × 3.6 × 0.93	30 × 25 × 1	4 × 4 × 30
Weight (g)	1	3	15

**Table 2 sensors-24-05853-t002:** AR(p) model parameters for Sensor 1.

	AR(1)	AR(2)	AR(3)	AR(4)	AR(5)
$\varphi_{1}$	0.8286	1.2538	1.4045	1.3892	1.3971
$\varphi_{2}$		−0.5131	−0.8815	−0.8353	−0.8020
$\varphi_{3}$			0.2938	0.2203	0.0940
$\varphi_{4}$				0.0523	0.2623
$\varphi_{5}$					−0.1512
AIC Values	1.0813	0.7719	0.6770	0.6756	0.6527

**Table 3 sensors-24-05853-t003:** AR(p) model parameters for Sensor 2.

	AR(1)	AR(2)	AR(3)	AR(4)	AR(5)
$\varphi_{1}$	0.8395	1.2690	1.3997	1.3712	1.3843
$\varphi_{2}$		−0.5116	−0.8359	−0.7426	−0.7309
$\varphi_{3}$			0.2556	0.0992	0.0119
$\varphi_{4}$				0.1117	0.2731
$\varphi_{5}$					−0.1177
AIC Values	1.1919	0.8892	0.8223	0.8105	0.7975

**Table 4 sensors-24-05853-t004:** Comparison of ME, MAE, and RMSE for two sensors.

	Criterion for Judgment	Original Signal	KF	Improved AR-SHAKF
Sensor1	ME	29.93	15.76	15.6
	MAE	7.40	6.88	6.86
	RMSE	8.75	7.52	7.46
Sensor2	ME	26.32	19.42	18.82
	MAE	7.09	6.61	6.57
	RMSE	8.53	7.51	7.43

**Table 5 sensors-24-05853-t005:** Comparison of sensor noise coefficients.

	Before Filtering		After Filtering	
	Sensor 1	Sensor 2	Sensor 1	Sensor 2
N	2.35	2.57	1.62	1.38
K	0.37	0.40	0.36	0.40
B	1.96	1.83	1.29	1.53

**Table 6 sensors-24-05853-t006:** Comparison table of ME, MAE, RMSE for two filters.

	Criterion for Judgment	Original Signal	KF	Improved AR-SHAKF
Sensor1	ME	26.71	26.33	25.98
	MAE	14.46	14.43	14.36
	RMSE	16.54	16.52	16.43
Sensor2	ME	4.97	4.96	4.95
	MAE	1.94	1.90	1.88
	RMSE	2.40	2.37	2.34

**Table 7 sensors-24-05853-t007:** Comparison of filtering effects.

	Original Signal	Improved AR-SHAKF	KF	FOLF	WRAF	$\varepsilon_{og-SHAKF}$	$\varepsilon_{og-KF}$	$\varepsilon_{og-FOLF}$	$\varepsilon_{og-WRAF}$
${\mathrm{ME}}^{\prime }$	0.0623	0.0461	0.0477	0.0491	0.0562	26.00%	23.43%	21.19%	9.79%
${\mathrm{MAE}}^{\prime }$	0.0118	0.0060	0.0082	0.0084	0.0099	49.15%	30.51%	28.81%	16.1%
${\mathrm{RMSE}}^{\prime }$	0.0159	0.0081	0.0117	0.0119	0.0136	49.06%	26.42%	25.16%	14.47%

**Table 8 sensors-24-05853-t008:** Comparison table of filtering effectiveness.

	Experiment 2 Signal	Error Rate (%)
${\mathrm{ME}}^{\prime \prime }$	0.0729	14.5
${\mathrm{MAE}}^{\prime \prime }$	0.0181	34.8
${\mathrm{RMSE}}^{\prime \prime }$	0.0237	32.9

## Data Availability

Due to commercial interests of the sensor manufacturer associated with the data, they are not disclosed.

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
