# Peer review of "Designing of Airspeed Measurement Method for UAVs Based on MEMS Pressure Sensors"

_sensors, 2024, doi:10.3390/s24175853_

Round 1

Reviewer 1 Report

Comments and Suggestions for Authors

This paper focuses on airspeed measurement for UAVs using MEMS pressure sensors. The approach aims to enhance the performance of these sensors, which are typically constrained by errors such as noise and bias due to their limited size.

The manuscript is well-written and presents its content effectively with excellent figures. Overall, it is well-organized.

Comments:

The results obtained from AR-SHAKF filtering are commendable when compared to the classic Kalman Filter, which is widely used in such applications. It is recommended to indicate the computational cost of both algorithms and specify the computational capabilities required for the microcontroller. This information is essential to determine the feasibility of implementing the entire algorithm on low-cost controllers (e.g., Arduino Uno, Arduino Nano, etc.).

Comments on the Quality of English Language

The quality of English language in the manuscript is generally strong, with clear and well-structured sentences that effectively convey the intended meaning. 

Author Response

Comments 1: The results obtained from AR-SHAKF filtering are commendable when compared to the classic Kalman Filter, which is widely used in such applications. It is recommended to indicate the computational cost of both algorithms and specify the computational capabilities required for the microcontroller. This information is essential to determine the feasibility of implementing the entire algorithm on low-cost controllers (e.g., Arduino Uno, Arduino Nano, etc.).

Response 1: Thank you very much for your valuable comments. The following parts are added to "Algorithm Complexity Analysis" in the article: ‘To compare the complexity of the KF and the Improved AR-SHAKF filtering algorithm proposed in this paper, we use the Big-O notation introduced by Juris Hartmanis and Richard E. Stearns to calculate complexity. In Big-O notation, O represents the order of magnitude, and O(f(n)) indicates how the complexity of the algorithm increases with the problem size n.

The complexity of the KF algorithm is mainly determined by the matrix operations in the prediction and update steps. The complexity of the prediction step is , and the complexity of the update step is . Typically, the state dimension n is greater than the observation dimension m, so  dominates the algorithm's complexity. Therefore, the complexity of the KF algorithm is .

The complexity of the Improved AR-SHAKF filtering algorithm proposed in this paper is divided into two parts: the AR model and the adaptive Kalman filter. The complexity of the autoregressive (AR) model is , and the complexity of the adaptive KF algorithm is . Therefore, the overall complexity of the Improved AR-SHAKF algorithm is . Similarly, since the complexity is dominated by , the complexity of the Improved AR-SHAKF algorithm is also . Thus, it can be seen that the complexities of the KF and Improved AR-SHAKF algorithms are similar.

The improved AR-SHAKF algorithm involves extensive matrix computations, requiring the microprocessor to have efficient computation units, support for single and double precision, and capabilities for parallel operations such as floating-point calculations. Additionally, it necessitates parallel processing capabilities like multi-core architecture and hardware threading, as well as sufficient memory bandwidth and capacity, including high-speed memory access, large memory capacity, and low memory latency.’

Reviewer 2 Report

Comments and Suggestions for Authors

The paper presents the design of a compact airspeed measurement device for small UAVs. A data processing method that combines mechanical filtering with the improved AR-SHAKF algorithm is proposed.

1. Why power consumption of the sensor is measured using uA ?

2. Figure 3 is tortured,check and fix it.

3. Sage-Husa Kalman Filte is widely used to deal with random noise, especially in MEMS devices. In the introduction part,authors introduced some researchers’ work,however,the cited literature is too old. Cite some articles published in the last two years to introduce the development status of SHKF in detail.

4. Adaptive Kalman filter was proposed in the sixties of the last century. The novelty of your algorithm,and their main contributions and superiorities are not clearly shown in the paper.

5. In the experimental part, sensors shown in figure 4 and figure 10 were different,why?Overall experimental setup should be introduced. How to control the speed of air flow in the experiment?

Comments on the Quality of English Language

The quality of English language is fine.

Author Response

Comments 1: Why power consumption of the sensor is measured using uA ?

Response 1: I'm sorry for the inconvenience caused. Due to my mistake, I wrote the sensor's power consumption as 3A, but it should be 3μA. I have made the correction in the section "Airspeed Measurement Device Structure" to “and power consumption as low as 3μA.”

Comments 2: Figure 3 is tortured,check and fix it.

Response 2: Thank you very much for your valuable feedback. I have updated the image accordingly.

Comments 3: Sage-Husa Kalman Filte is widely used to deal with random noise, especially in MEMS devices. In the introduction part,authors introduced some researchers work,however,the cited literature is too old. Cite some articles published in the last two years to introduce the development status of SHKF in detail.

Response 3: Thank you very much for your valuable feedback. I have included the following information in the document: The literature [32] proposes a Sage–Husa adaptive Kalman filtering-based pedestrian characteristic parameter update mechanism, which enhances step length estimation in pedestrian dead reckoning, resulting in improved accuracy. The literature [33] presents an improved Sage–Husa Adaptive Robust Kalman Filtering (MSHARKF) algorithm, which reduces random error noise to one ten-thousandth and one hundredth of the original data, respectively. In summary, the main issue with the Sage-Husa adaptive filter algorithm, as evident from the above references, is filter divergence caused by the uncertainty in filter noise.

Comments 4: Adaptive Kalman filter was proposed in the sixties of the last century. The novelty of your algorithm,and their main contributions and superiorities are not clearly shown in the paper.

Response 4: Thank you very much for your valuable feedback. Please include the following additions in the introduction and conclusion sections of the document:

Finally, wind tunnel tests show that the airspeed measurement device designed in this paper, along with the proposed adaptive AR-SHAKF algorithm, achieves significantly lower ME, MAE, and RMSE compared to the KF, demonstrating a notable improvement in noise reduction and more accurate airspeed calculation results.

This paper designs a compact airspeed measurement device for small UAVs. Given the small size and low cost of these UAVs, a data processing method combining mechanical filtering and an improved AR-SHAKF algorithm is proposed. The mechanical filtering reduces the impact of the UAV's body on the airflow by adding a metal tube. The improved AR-SHAKF algorithm establishes a state equation using an autoregressive model based on static sensor data, determines initial noise values, and restricts the measurement noise matrix range to achieve adaptive noise estimation, preventing filter divergence. The algorithm shows significantly lower ME, MAE, and RMSE compared to KF, resulting in more accurate airspeed calculations.

Comments 5: In the experimental part, sensors shown in figure 4 and figure 10 were different,why?Overall experimental setup should be introduced. How to control the speed of air flow in the experiment?

Response 5: The sensor testing in Figure 4 is a laboratory static test used to measure the autoregressive (AR) model of the pressure sensor. The sensor in Figure 10 represents wind tunnel test data, which is used to evaluate the accuracy of the airspeed measurement device. Additionally, include the following description in the document: The wind speed in the wind tunnel is controlled using a pressure control valve with closed-loop regulation. The following section has been added to the document: Before conducting the wind tunnel tests, the wind speed and duration of the wind tunnel are set using a pressure control valve with closed-loop control. This value is then plotted over time to represent the true airspeed measurement.

Reviewer 3 Report

Comments and Suggestions for Authors

Measuring airspeed is essential for controlling UAVs. To obtain precise airspeed readings, this paper calculates the airspeed by monitoring variations in air pressure and temperature. Building on this approach, a data processing technique that combines mechanical filtering with an enhanced AR-SHAKF algorithm is introduced to achieve high-precision indirect airspeed measurement. Specifically, a mathematical model for the airspeed measurement system is developed, and a method for installing the pressure sensor is designed to measure total pressure, static pressure, and temperature. The measurement principles of the sensor are then analyzed, and a metal tube is implemented as a mechanical filter, especially to mitigate the effects of significant disturbances in the gas flow field caused by the aircraft. The results show that the airspeed measurement device and the developed velocity measurement method can effectively calculate airspeed with high accuracy and strong resistance to interference. The paper is logically complete and well-structured, with a thorough modeling, analysis, and testing process. It has significant research importance.

1.     It is suggested to include a flowchart at the beginning to help readers clearly understand the overall design structure of the paper.

2.     The experimental section could be enhanced by including an analysis of the estimator's sensitivity to noise.

3.     The conclusion of the paper should address the limitations of the proposed method and suggest directions for future research

4.     In addition to UAVs, the literature review section should also highlight the significance of vehicle speed measurement for autonomous driving systems, as illustrated in Reference “A Robust Dynamic Game-Based Control Framework for Integrated Torque Vectoring and Active Front-Wheel Steering System, IEEE Transactions on Intelligent Transportation Systems, vol. 24, no. 7, pp. 7328-7341, July 2023”.

Comments on the Quality of English Language

Minor editing of English language required.

Author Response

Comments 1:It is suggested to include a flowchart at the beginning to help readers clearly understand the overall design structure of the paper.

Response 1: Thank you very much for your valuable feedback. The following section has been added to the document: The process of this paper is shown in Figure 1, and the structure is as follows.

Figure 1. The airspeed measurement flowchart.

Comments 2: The experimental section could be enhanced by including an analysis of the estimator's sensitivity to noise.

Response2:  Thank you very much for your valuable feedback. In the document, the AR-SHAKF method proposed in this paper is compared with the first-order lag filtering method presented in the literature.

To analyze the advantages of the improved AR-SHAKF method proposed in this paper, we compare the airspeed values calculated using this method with those obtained from other methods. In reference, a first-order lag filter is employed to optimize the pressure values measured by the sensor. The filtering equation is as follows:

In the equation,  represents the current filtering output value,  represents the current readout data,  represents the previous filtering output value, and k is a filtering coefficient ranging from 0 to 1.

In reference [Design and implementation of barometric height measurement system], based on the recursive averaging filtering algorithm, N data points are allocated with weighting coefficients to form a weighted recursive averaging filtering (WRAF) algorithm, as shown below:

In the equation,  represents the current filtering output value, ,  and  represent the sensor measurement values at times n, n-1, and n-2 respectively.  ,  and  are the weighting coefficients satisfying .

The airspeed measurements are processed for noise reduction using the improved AR-SHAKF filter, KF, FOLF, and WRAF. The filtered data is then compared with the true airspeed values, as depicted in the figure 14.

Figure 14: Comparison of Airspeed Calculated Results with True Airspeed Values

Table 6: Comparison of Filtering Effects

Original

signal

Improved AR-SHAKF

KF

FOLF

WRAF

(%)

(%)

(%)

(%)

0.0623

0.0461

0.0477

0.0491

0.0562

26.00

23.43

21.19

9.79

0.0118

0.0060

0.0082

0.0084

0.0099

49.15

30.51

28.81

16.1

0.0159

0.0081

0.0117

0.0119

0.0136

49.06

26.42

25.16

14.47

Comparing the ME, MAE, and RMSE of the filtered signals with the true signal values reveals that the error is smallest for the signal filtered using the improved AR-SHAKF filter, followed by the KF. The FOLF and WRAF exhibit the largest errors. The improved AR-SHAKF filter demonstrates the greatest reduction in airspeed value error after noise reduction, resulting in airspeed values that are closer to the true values. Among the four filters, the filtering effect of the improved AR-SHAKF filter is optimal, providing the best noise removal performance.

Comments 3: The conclusion of the paper should address the limitations of the proposed method and suggest directions for future research

Response3: Thank you very much for your valuable feedback. The following content has been added to the conclusion: However, the proposed methods require high hardware computational resources, and further research will be conducted to address this issue.

Comments 4: In addition to UAVs, the literature review section should also highlight the significance of vehicle speed measurement for autonomous driving systems, as illustrated in Reference “A Robust Dynamic Game-Based Control Framework for Integrated Torque Vectoring and Active Front-Wheel Steering System, IEEE Transactions on Intelligent Transportation Systems, vol. 24, no. 7, pp. 7328-7341, July 2023”.

Response4: Thank you very much for your valuable feedback. The following content has been added to the document: In [17] proposes an integrated control framework of Torque Vectoring (TV) and Active Front Steering (AFS) systems to ensure the stability of the vehicle's lateral dynamics.

Round 2

Reviewer 3 Report

Comments and Suggestions for Authors

The authors have made a commendable effort to address most of my concerns within the available space. The revised version of the manuscript is enhanced noticeably. I would recommend this paper to be published. Please standardize the format of the references before publishing. Furthermore, the literature background should further remark the significance of speed measurement for autonomous driving systems, “An Energy-Oriented Torque-Vector Control Framework for Distributed Drive Electric Vehicles, IEEE Transactions on Transportation Electrification, vol. 9, no. 3, pp. 4014-4031, Sept. 2023”.

Author Response

Comments 1:Furthermore, the literature background should further remark the significance of speed measurement for autonomous driving systems, “An Energy-Oriented Torque-Vector Control Framework for Distributed Drive Electric Vehicles, IEEE Transactions on Transportation Electrification, vol. 9, no. 3, pp. 4014-4031, Sept. 2023”.

Response 1: Thank you very much for your valuable feedback. The following section has been added to the document: The paper [] proposes a hierarchical framework based on dual-model predictive control (MPC) to achieve energy savings while enhancing the processing stability of DDEV.
